# Long-Term Forecast of Sierra Leone's Energy Supply and Demand (2019–2040): A LEAP Model Application for Sustainable Power Generation System

Foday Conteh [1,*], Masahiro Furukakoi [2], Shriram Srinivasarangan Rangarajan [3,4],
Edward Randolph Collins [4,5], Michael A. Conteh [6], Ahmed Rashwan [7] and Tomonobu Senjyu [1,*]

[1] Department of Electrical and Electronics Engineering, University of the Ryukyus, Okinawa 903-0213, Japan
[2] National Institute of Technology, Sasebo College, Sasebo 857-1193, Japan; e125511@gmail.com
[3] Enerzinx India Private Limited, Velankani Tech Park, No. 43, 3rd Floor South Wing, Block 1, Hosur Rd, 6 Suryanagar Phase I, Electronic City, Bengaluru Karnataka 560100, India; shriras@g.clemson.edu
[4] Department of Electrical and Computer Engineering, Clemson University, Clemson, SC 29634, USA; rcollins@wcu.edu
[5] College of Engineering, Western Carolina University, Cullowhee, NC 28723, USA
[6] Mechanical Engineering Department, Faculty of Engineering and Architecture, Fourah Bay College, University of Sierra Leone, Mount Aureol, Freetown P.O. Box 87, Sierra Leone; maconteh1865@gmail.com
[7] Electrical Networks Department, Faculty of Energy Engineering, Aswan University, Sahary City, Aswan P.O. Box 81528, Egypt; engrashwan@aswu.edu.eg
* Correspondence: contehfoday1979@gmail.com (F.C.); b985542@tec.u-ryukyu.ac.jp (T.S.)

**Abstract:** Sierra Leone is suffering from a persistent electricity gap that has crippled its economic growth and prevented it from attaining several health and education development goals. This persistent electricity gap has generated significant interest in tackling the country's long-lasting energy deficiency. Providing electricity in a reliable, sustainable, and cost-effective manner in Sierra Leone requires adopting robust integrated energy planning and appropriate technologies. Despite various interventions by the government, a balance between electricity demand and supply has yet to be achieved. Using the Long-range Energy Alternatives Planning System (LEAP), this work assesses Sierra Leone's energy supply and demand for 2019–2040. We developed three case scenarios (Base, Middle, and High) based on forecasted demand, resource potential, techno-economic parameters, and $CO_2$ emissions. The Base case considers the electricity sector as business as usual, the Middle case examines the electricity sector reform roadmap and the prospect of integrating renewable energy into the power system, and the High case examines the sustainable development of the power generation system considering the electricity sector roadmap. As part of this study, we analyze potential alternatives to conventional electricity generation systems aimed at providing electricity in a sustainable, reliable, and affordable manner, including the use of renewable energy sources and technologies with less $CO_2$ emissions. Model results estimate an increase in electricity demand of 1812.5 GWh, 1936 GWh, and 2635.8 GWh for Base, Middle, and High cases respectively. Also, there is a reduction in production, fuel cost, and $CO_2$ emission in the High case to the Base case by 67.15%, 35.79%, and 51.8%, respectively. This paper concludes with recommendations devised from the study results for the power system of Sierra Leone.

**Keywords:** electricity forecasting; GHG emission; LEAP; scenario analysis

## 1. Introduction

Approximately 1.2 billion people live without electricity, with Sub-Saharan Africa and South Asia having the most significant deficit [1]. There is a vast gap between demand and supply in Africa. The existing infrastructure needs to be improved to meet current demand and the growth of the coming decades. In Sub-Saharan Africa, the installed capacity was about 90 GW in 2012, expected to rise to 380 GW in 2040 [2]. The continent

is home to more than 950 million people and is the most electricity-poor region globally. More than 600 million people lack electricity access, and many are connected to an unreliable grid that does not meet their daily energy service needs [3]. West Africa, to which Sierra Leone belongs, holds one of the largest populations on the continent. According to a 2017 report, only 51.3% of the region's population has access to electricity, indicating that about 170 million people in the area are without electricity. Even those with access to electricity suffered from frequent power cuts [4]. Sierra Leone is troubled by a complex and persistent electricity gap. Closing the electricity gap in Sierra Leone remains a considerable challenge with imperative implications. Approximately 23% of the Sierra Leone population has access to electricity [5]. 60% of Sierra Leone's population lives in rural areas. Persistent electricity scarcity has crippled the country's economic growth and prevented it from attaining several health and education development goals. This is mainly a result of inadequate generation capacity to supply power to grid-connected regions, the absence of proper grid infrastructure to deliver this power, and regulatory impediments to providing a steady revenue to maintain and invest in new generation capacity. Sierra Leone has consistently ranked in the bottom ten of all African countries regarding infrastructure development, primarily due to poor service provision in the energy sector [6]. The country's infrastructure is still suffering from the consequences of physical damage incurred during the country's eleven-year civil war. In 2002, Sierra Leone's civil war ended, leaving tens of thousands dead and millions of displaced people and having destroyed much of the energy infrastructure. Nearly half of Sierra Leone's working-age population engages in subsistence agriculture and the country possesses substantial mineral, agricultural, and fishery resources. Mining—particularly iron ore—has recently driven economic growth and poverty reduction. However, these gains were severely eroded by the outbreak of the Ebola Virus Disease in 2014 and its ensuing consequences, as well as the fall in the price of iron ore in the world market, impacting the country's gains from its key export material [7]. The country's principal exports are iron ore, diamonds, rutile, and gold, and the economy is vulnerable to fluctuations in international prices. Until 2014, the government had relied on external assistance to support its budget, but it was gradually becoming more independent [8]. The Ebola outbreak of 2014 and 2015, combined with falling global commodities prices, caused a significant contraction of economic activity in all areas. The advent of the COVID-19 pandemic has lately exacerbated this situation. Increasing generation and improving access to electricity through transmission and distribution network expansion remains a top priority for the Government of Sierra Leone (GoSL). To address the issues mentioned above, the GoSL, over recent years and with support from the Economic Community of West African States Centre for Renewable Energy and Energy Efficiency (ECREEE), Millennium Challenge Corporation (MCC), Japan International Cooperation Agency (JICA), and other development partners, prepared several policy papers and proposals aimed at reforming the energy sector. The reform sector project (RSP) creates the establishment of the electricity reform sector project (ERSP), electricity sector expansion planning, and electricity sector asset inventory and revaluation [9]. Under the ERSP, the vertically integrated national power authority (NPA) was unbundled into the electricity generation and transmission company (EGTC) and electricity distribution and supply authority (EDSA) through an act of Parliament enacted in 2011. A regulatory body, the electricity and water regulatory commission (EWRC), was also set up to oversee the affairs of the two entities. The ERSP aims to address the policy and institutional constraints associated with inadequate, unreliable, and unaffordable access to electricity by providing technical assistance and coordination to EGTC, EDSA, EWRC, and the Ministry of Energy (MoE). The ERSP consists of two primary activities:

- The electricity sector reform roadmap and Coordination Activity
- Institutional strengthening activity

The electricity sector reform roadmap and coordination activity aim to establish a sector steering committee and a roadmap that will clarify the role of sector stakeholders, including financial contributions from the GoSL and independent power producers (IPP).

The roadmap would improve coordination and provide an open tender for increased electricity generation. Institutional strengthening activity focuses on technical assistance, primarily to EGTC and EDSA, to enhance coordination between the two entities and provide information for improved operations and decision-making, all to reduce the cost of service through increased financial and technical capacity of the sector [10]. In developing the electricity sector reform roadmap, the GoSL emphasized the following priorities:

1.  Develop electricity supply public service where the public and economy need it most;
2.  Embrace partnerships with the private sector to create a sustainable and inclusive electricity supply public service and
3.  Focus on the complex needs of Sierra Leone's population and business community at the national and local levels. In 2018, the GoSL allocated 15.6 million USD to the energy sector to implement the electricity sector reform roadmap.

This was geared towards increasing electricity generation and rebuilding and enhancing the distribution network. In support of this initiative, development partners, such as the World Bank and JICA, pledged 43.7 million USD for various energy projects. The United States government, through the Millennium Challenge Coordination (MCC) in 2015, granted 44.4 million USD to the GoSL in support of several reform programs. One of these programs is geared toward electricity services in Sierra Leone [11]. The Millennium Challenge Coordinating Unit (MCCU) manages the funds from MCC, an entity established by Parliament with oversight functions carried out by the MCCU Board, chaired by the Vice President of the Republic of Sierra Leone. MCCU collaborates with EGTC and EDSA by providing institutional strengthening support and capacity building through consultancy services. Sierra Leone lies on the west coast of Africa. Guinea borders it in the northeast, Liberia in the southeast, and the Atlantic Ocean in the west. The country has a total land area of 71,740 km$^2$ with a population of approximately 7,534,981 people [12]. The country's gross domestic product (GDP) is 4.2 billion USD [13]. Sierra Leone has a relatively small economy, and its economic growth is volatile, tiny, and undiversified. The country's economy dropped by 3.7 percent in 2017, compared to 6.3 percent in the previous year. This was primarily due to the weak recovery of mineral production, especially iron ore [14]. The rise in fuel prices also resulted in widespread power outages. The GDP growth estimate for 2022 has been lowered from 5% to 3.6%. However, the economy is expected to grow due to the resumption of iron-ore mining at Marampa and Tonkolili, and the gradual recovery in electricity consumption. [15]. Growing demand for electricity is expected to increase due to an increase in population, the restart of the mining sector, and the rapid urbanization of towns and villages.

There has been no proper energy demand forecasting study in Sierra Leone for the past decade. However, energy demand forecasting for short, medium, and long-term planning has been carried out by many researchers. Mirjat et al. propose a long-term electricity demand forecast and supply-side scenarios for Pakistan (2015–2050), utilizing a LEAP Model application for policy analysis [16]. Four supply-side scenarios were presented in the article, including Reference (REF), Renewable Energy Technologies (RET), Clean Coal Maximum (CCM), and Energy Efficiency and Conservation (EEC), to estimate the demand forecast for 2050, assuming an annual average growth rate of 8.35%. However, the author found that the RET scenario was the most sustainable electricity generation path followed by the EEC scenario, despite being capital-intensive earlier in the modeling period. Nieves et al. worked on energy demand and greenhouse gas emissions analysis in Colombia using the LEAP model application [17]. An analysis of Colombia's energy demand and greenhouse gas emissions was proposed by the authors. In order to predict the future of the country's economic sectors (tertiary, industrial, housing, and transport), the author uses the LEAP model for 2015 and extrapolates it to 2030 and 2050. There were two scenarios implemented (positive and negative). As a result of low economic growth and few incentives for technological change, the negative scenario is characterized by low energy efficiency growth and low petroleum energy substitution. Positive scenarios, however, have higher economic growth and faster technological substitution. This results

in higher energy efficiency levels in energy final use and an increased migration towards cleaner technologies. Kale et al. propose electricity demand and supply scenarios for Maharashtra (India) for 2030 using LEAP software [18]. For Maharashtra, the authors used the LEAP model to forecast electricity demand for 2030. The author utilizes Holt's exponential smoothing method to arrive at suitable growth rates. Three scenarios, Business as Usual, Energy Conservation, and Renewable Energy, were generated. Analysis based on energy, environmental impact, and cost were also investigated. Model results indicated that projected electricity demand for business as usual and renewable energy increased by 107.3 percent for the target year 2030 over the base year 2012 and electricity demand has grown by 54.3 percent. The estimated values of greenhouse gases for business as usual and energy conservation in the year 2030 are 245.2 percent and 152.4 percent more, respectively, than the base year; for renewable energy, this value is 46.2 percent. Yophy et al. conducted a long-term forecast of Taiwan's energy supply and demand using the LEAP model application [19]. The authors conducted an overview of energy supply and demand in Taiwan and a summary of the historical evolution and current status of its energy policies through the application of the LEAP model. The authors used the LEAP model to compare future energy demand and supply patterns as well as greenhouse gas emissions for several alternative scenarios. Various scenarios, including Business as usual policies, Aggressive energy-efficiency improvement policies, and On-schedule retirement of Taiwan's three existing nuclear plants were provided and compared, along with sensitivity cases exploring the impacts of lower economic growth assumptions.

Despite numerous strides made by the government of Sierra Leone to improve the electricity access rate, progress has been slow, especially in rural areas where 60% of the population resides. Because of inadequate planning, Sierra Leone is struggling to implement the objectives of the ERSP. This work proposes integrated resource planning (IRP) that assesses the country's current and future electricity supply and demand. The rationale behind this IRP is to ensure that generation meets the need, enhances reliability, minimizes the total cost, and reduces environmental costs and greenhouse gas emissions (GHG). As such, IRP has become one of the most effective tools for meeting energy demand [20]. Sierra Leone needs appropriate IRP and policy frameworks to meet its current and future energy demands. With support from MCCU in 2022, AF Mercados prepared an Integrated Resource Plan for Sierra Leone to support the electricity reform sector roadmap to meet the country's objective of providing reliable and least-cost electric service to all customers while considering substantial risks and uncertainties inherent in the electric utility business [21]. This work proposes three different scenarios using the LEAP software to address current and future electricity demand. The demand forecasts for 2019 to 2040 were simulated using three potential demand scenarios: Base case, Middle case, and High case. The work will help policymakers in the electricity sector by providing the necessary information to meet current and future electricity demand. It will also address contributing factors that impact electricity supply and delivery. Section 2 of this work analyzes of the power sector of Sierra Leone. The LEAP methodology and scenario development, as well as the model results relating to demand and supply projections, are analyzed in Section 3. Cost and sensitivity analyses and GHG emissions drawn on for these three scenarios are discussed in Section 4. Section 5 discusses the results while the conclusions are given in Section 6.

## 2. Electricity Situation in Sierra Leone

Sierra Leone is troubled by a complex and persistent electricity gap. Closing the electricity gap in Sierra Leone is a multidimensional challenge with significant implications. Having access to electricity helps people and communities increase their incomes and productivity, enhance access to health care, water, and education, and improves their overall well-being [22]. Sierra Leone's generating capacity is about half that of Togo and 1/200th that of Switzerland, countries with similarly-sized populations [23]. The country's publicly owned power sector currently has approximately 116.81 MW installed capacity and roughly 255,993 grid-connected customers as of 2021. This installed capacity does not

include the IPPs (65 MW from the Turkish Karpowership and 27 MW from CLSG). The current electricity supply network of Sierra Leone comprises a 204 km 161 kV single circuit line that runs from the Bumbuna hydropower plant in the north to the capital city Freetown and a 33 kV line connecting the eastern and southern towns of Kenema and Bo, respectively. The country's most extensive transmission project, a 1300 km, 225-kV interconnection between Cote D'Ivoire, Liberia, Sierra Leone, and Guinea (CLSG), passes through the south, east, and north of the country. Five substations are allocated in Sierra Leone, of which three have already been completed. One of the completed substations serves the eastern and southern cities of Kenema and Bo, respectively. The country's electricity supply and demand gap is so vast that only 23% of the country's population of 7,534,981 have access to electricity; this is below the Sub-Saharan average of 30%. This supply and demand gap affects people's welfare and ability to access services and impedes the country's competitiveness in global activities, job creation, and poverty reduction [24]. In addition to this constraint, high transmission and distribution losses represent an estimated 38% of total electricity generated. Major industries, including the mining sector, rely on private generators for their energy needs, costing them millions of Leones daily. This complex situation prompted the government of Sierra Leone to negotiate with a floating Turkish power ship called Karpowership in 2018, anchored off the coast of the capital city Freetown, to supply 65 MW of electricity at the cost of approximately USD 2 million per month [25]. Unfortunately, EDSA, which is the off-taker, has been struggling to honor the monthly payment of energy bills sent by Karpowership, resulting in the government owing Karpowership approximately USD 36 million. The accumulating arrears from the government prompted Karpowership to ration the power supply [26]. The lack of adequate transmission and distribution lines contributed significantly to the difficulty of accessing electricity. Only a few people enjoyed electricity from the grid, and those people faced frequent power cuts. Despite the many challenges facing the energy sector, the government has demonstrated its commitment to enhancing energy access through the enactment of the ESRR and the Sierra Leone Sustainable Energy for All (SE4ALL) Country Action Agenda. The ESRR defined the responsibilities of the various entities: EWRC is responsible for regulating and setting consumer tariffs and tariffs between IPPs and EGTCs; EGTC is responsible for electricity generation and transmission of electricity; EDSA has the sole responsibility for the distribution of this electricity to the various consumers. The SE4ALL Country Action Agenda aims to increase electricity access from 13% in 2013 to 92% by 2030, and the renewable energy level from 43,464 GWh in 2013 to 111,780 GWh by 2030 [27]. To achieve this goal, the GoSL allocated USD 15.6 million from the domestic budget for electricity generation, thermal plant upgrades, and distribution network improvements. As part of this initiative, development partners have also pledged USD 43.7 million toward various energy projects. In 2015, the US government (through the MCC) granted the GoSL up to USD 44.4 million over a 4-year period to support several reforms, including the provision of electricity. A USD 40 million agreement was also signed by the International Finance Corporation (IFC) in October 2018 to build a 50 MW solar power plant in Sierra Leone.

In addition, the GoSL, with support from the ECOWAS Bank for Investment and Development (EBID), launched the project for the rural electrification of seven districts' headquarters towns, including Kambia, Kabala, Kailahun, Bonthe, Moyamba, Pujehun, and Matru-Jong [28], as illustrated in Figure 1. Table 1 shows forecasted generation from various generation technologies for 2030 by MoE.

### 2.1. Electricity Supply Situation

As of 2018, Sierra Leone's largest generation source is a pair of barge-mounted heavy fuel power plants off the coast of Freetown (Karpowership) that jointly supply 65 MW of capacity during the dry season and 23 MW during the rainy season in the capital Freetown. The other primary generating source is the state-owned Bumbuna hydroelectric plant, which provides 50 MW during the rainy season and 8 MW during the dry season. Approx-

imately 90 percent of the power produced from the Bumbuna hydroelectric dam gets to Freetown, while the city of Makeni in the north, located along the 161 kV line, receives power from the shield wires. Two smaller hydropower plants, Charlotte and Bankasoka, produce 2 MW and 2.2 MW, respectively. However, these smaller hydropower plants are forced to shut down completely during the dry season due to the low level of water. Sunbird Bioenergy, a company that produces bio-ethanol from sugarcane plantations, uses its residues to generate 32 MW of electricity, of which up to 15 MW is injected into the national grid [29]. In addition, the regional interconnection TRANSCO CLSG 225 kV line, which runs from Cote D'Ivoire to Sierra Leone, began to supply 27 MW to Sierra Leone in December 2021 through a power purchase agreement signed between the two countries [30]. This power is currently distributed to Bo, Kenema, and the capital, Freetown. Twenty-nine rural communities within 3 km alongside the CLSG interconnection transmission line are to be supplied with electricity, which will subsequently improve the livelihood of the greater population living in these communities [31]. Table 2 presents the country's current generation facilities. Total electricity generation by hydro, fuel oil thermal plants, and solar in 2021 was 229,844,780, 325,895,860, and 4,500,000, respectively. This represents 41% for hydro, 58% for fuel oil thermal plants, and 1% for solar, as seen in Figure 2. Generation installed capacity from renewable (Hydro and Solar) has increased from 50 MW in 2015 to 66.52 MW in 2021, representing a 33.04% increase, and that from fuel oil thermal plants increased from 43.4 MW in 2015 to 179.4 MW in 2021, representing a 313.36% increase. This increase in generation includes the 65 MW from the Karpowership. Total gross electricity generated increased from 247,047.287 MWh in 2015 to 551,240.640 MWh in 2021, representing a 123.13% increase, as indicated in Table 3. Sierra Leone lacks a stable and reliable public power supply, and domestic demand remains significantly unmet. Some of the contributing factors to these problems are the lack of adequate finances to invest in the energy sector and limited private sector participation. Other causes include ineffective planning and applicable policy to ensure sustainable electricity supply.

**Table 1.** 2030 Forecasted Generation for Various Generation Technology (MoE).

| Generation Technology | Generation in 2030 (MW) |
|---|---|
| Thermal | 90 |
| Large- and Small-Scale Hydro | 560 |
| Renewable | 120 |
| Total | 770 |

**Table 2.** Sierra Leone's current electricity generation facilities in megawatts (MW).

| Generation Location | Region | Technology | Nominal Power |
|---|---|---|---|
| Bumbuna | North | Hydro | 50 |
| Dodo | East | Hydro | 6 |
| Bankasoka | Hydro | North | 2 |
| Charlotte | Hydro | West | 2.2 |
| Makali | Hydro | North | 0.32 |
| Kingtom | HFO | West | 10 |
| Blackhall Road | HFO | West | 16.4 |
| Lungi | HFO | Northwest | 6 |
| Kono | HFO | East | 6 |
| Bo | DFO | South | 4 |
| Makeni | DFO | North | 6.08 |
| Magburaka | DFO | North | 0.8 |
| Lunsar | DFO | North | 1 |
| Newton | Solar | West | 6 |
| Karpowership | HFO | West | 65 |
| CLSG | | | 27 |
| Total | | | 208.81 |

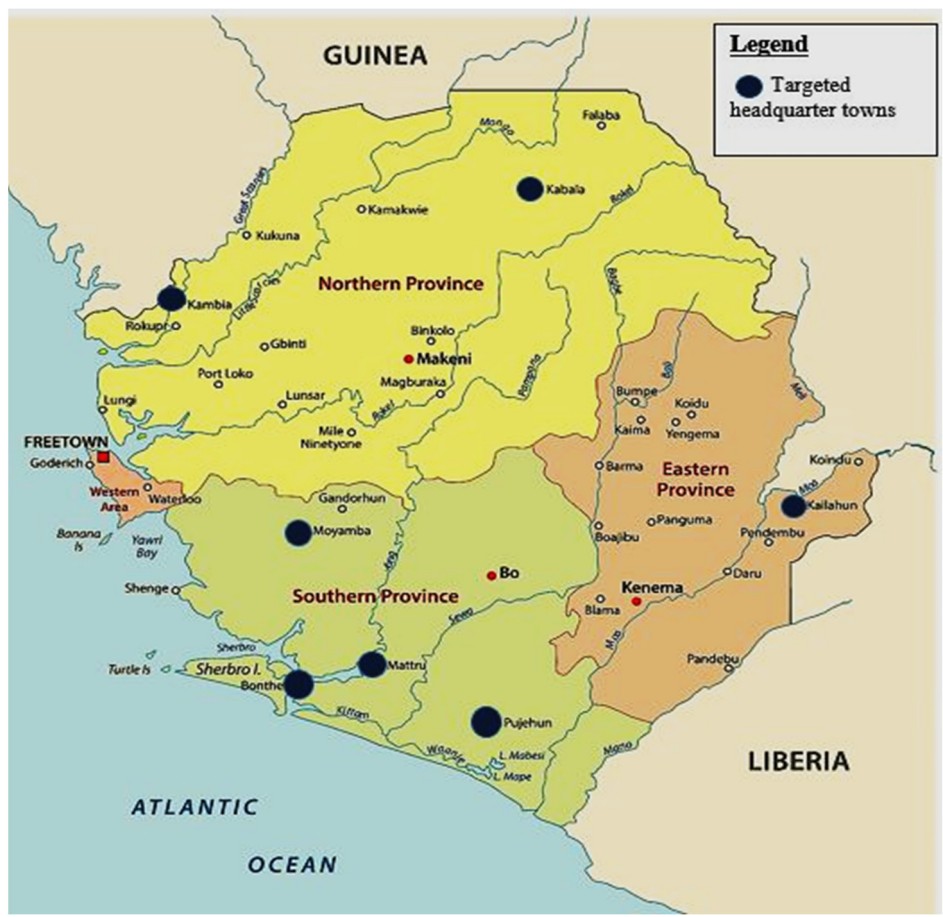

**Figure 1.** Map of Sierra Leone indicating the seven district headquarter towns for the rural electrification project.

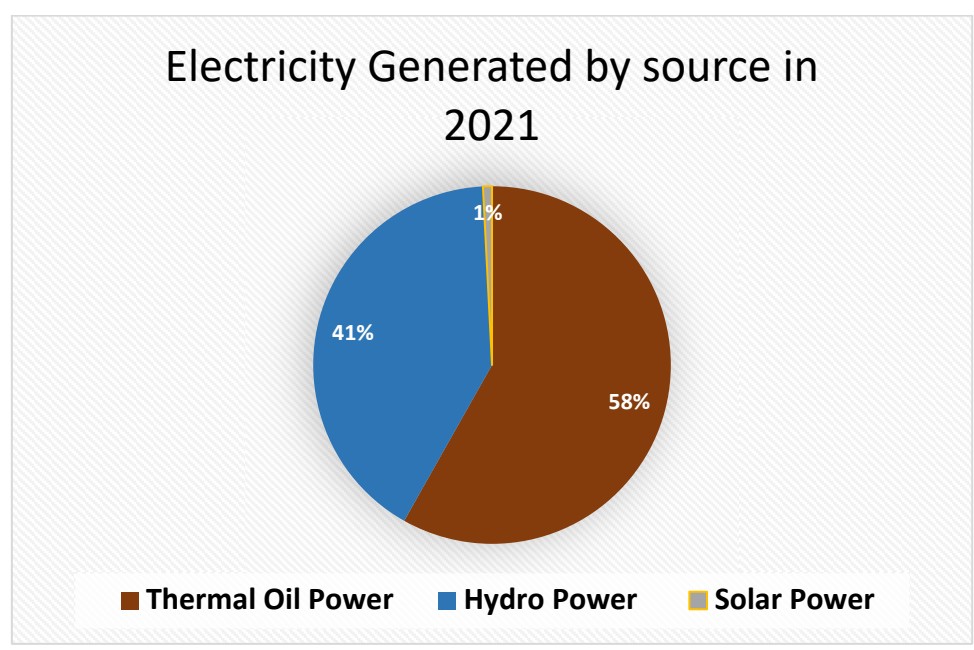

**Figure 2.** Percentage Electricity Generation by Source in 2021.

**Table 3.** Electricity production by source for 2015 and 2021 (MWh).

| Year | Total Energy Generated |
|------|------------------------|
| 2015 | 247,045.286 |
| 2016 | 310,301.415 |
| 2017 | 317,162.206 |
| 2018 | 387,646.622 |
| 2019 | 403,920.524 |
| 2020 | 442,244.437 |
| 2021 | 551,240.640 |

*2.2. Electricity Demand Situation*

The rapid urbanization of towns and villages, along with the growth of the mining sector, are believed to contribute to the increase in electricity demand. In 2015, the country's electricity demand was 52.4 MW, with the capital city Freetown having a demand of 42.4 MW and 10 MW for provincial areas. The country's electricity demand increased to 106.91 MW in 2019, with Freetown having a demand of 82 MW and 24.91 MW for the province. Figure 3 shows the electricity demand and supply gap for 2015 to 2021, with frequent load shedding to prevent system cascades.

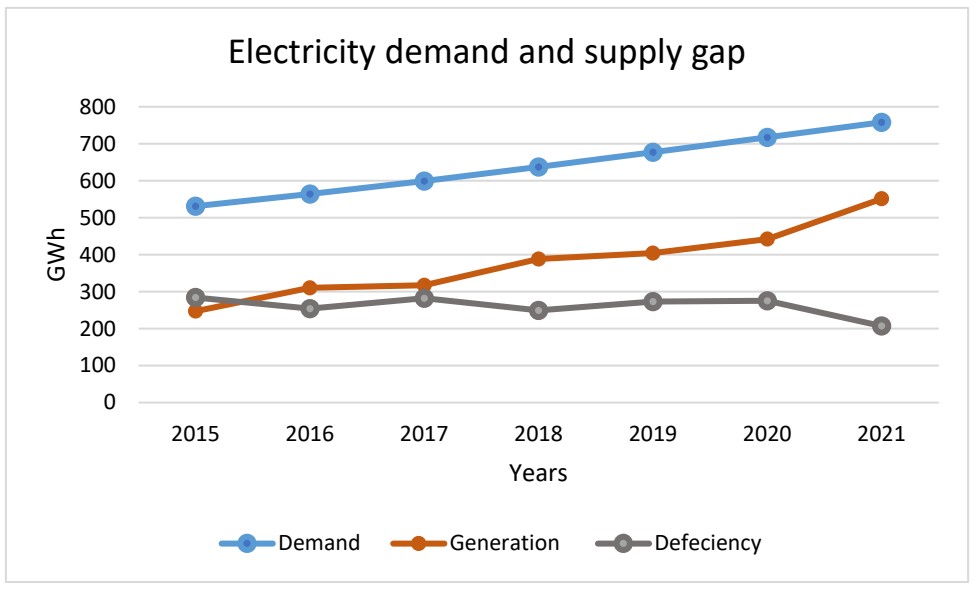

**Figure 3.** Electricity demand and supply gap.

Electricity customers increased from 103,664 in 2015 to 255,993 in 2021, representing a 118% increase as shown in Table 4. Electricity consumption in the country is divided into four major categories: industry, commercial, household, and agriculture. Industry and commercial consume about 61% of the electricity generated, households about 37%, and approximately 2% in agriculture [32]. However, the country's electricity demand outweighs the generation, resulting in frequent load shedding across the country. Even though there has been an increase in the consumption of electricity over recent years, the electricity consumption per capita currently stands at 55 kWh, placing it as one of the lowest in the world. The country's transmission and distribution infrastructure is limited, with most parts having no grid access [33]. For Sierra Leone to meet future power consumption, which looks at both the demand side (conservation, energy efficiency, etc.) and supply side (generation/power plants, transmission lines, etc.), as well as reducing sector inefficiencies and electricity costs, integrated resources planning (IRP) is required to meet the future electric energy needs of the country.

**Table 4.** EDSA Customer Base for 2015 to 2021.

| Year | Total Number of Customers |
|------|---------------------------|
| 2015 | 103,664 |
| 2016 | 130,776 |
| 2017 | 159,888 |
| 2018 | 184,997 |
| 2019 | 202,291 |
| 2020 | 229,158 |
| 2021 | 255,993 |

## 3. Methodology

The LEAP model was chosen since it is a scenario-based, integrated tool that tracks of energy consumption, production, and resource extraction across all sectors [17]. It accounts for both energy and non-energy sectors and greenhouse gas emission sources and sinks. The model structure has three primary programs or modules:

- Demand module at the top
- Transformation module midway
- Resources module at the bottom

It is possible to evaluate energy systems at different scales, including geographically (e.g., cities, states, countries, regions, or global) and for other aggregation levels (e.g., project, single-sector, multi-sectors, and multi-regional) [34]. To determine the total energy demand and consumption, LEAP examines the product of the departmental activities and the level of energy intensity within them. The activities of the department include household, commercial, industrial, and agricultural [35]. Equation (1) shows the exact formula for total energy consumption.

$$EC = \Sigma\Sigma\ AL_{nji}.EI_{nji} \tag{1}$$

where *EC* is the total energy consumption for a given sector, *AL* represents activity level, *EI* is the energy intensity, *n* is the fuel type, *I* is the sector, and *j* is the device [36].

As part of the atmospheric pollution analysis process, the Technology and Environmental Database (TED) method-based emission per consumed energy unit for each fuel source is used. Pollution emissions are classified hierarchically according to energy demand by system sectors [34]. Equation (2) calculates the carbon emissions of the final energy.

$$CEC\ =\ \Sigma\Sigma\Sigma_{j}\ A_{nji}.E_{nji}.EF_{nji} \tag{2}$$

where *CEC* is the carbon emission, *EF* is the carbon emission factor, *AL* represents activity level, *EI* is the energy intensity, n is the fuel type, *I* is the sector, and *j* is the device.

The model has various modules, including key assumptions, demand, transformation, resources. etc. In the key assumption module, data such as total population, urban and rural population, households, GDP, and similar data have been used. Demand analysis is used for modeling final energy consumption requirements. In the demand module, this consists of four sectors: household, industrial, commercial, and agricultural. The industrial sector is subdivided into two, including the production and mining industries. Generally, the methodology used involves the construction of three projection scenarios: Base, Middle, and High cases for each sector. Energy demand projections were performed for 2019 to 2040. The electricity consumption values for 2019 have been utilized for forecasting electricity demand and supply, predicting GHG emissions, and cost analysis.

### 3.1. Sierra Leone LEAP Model

LEAP software analyzes long-term electrical demand and supply forecasts. The period under study ranges from 2019 to 2040, with statistical data from 2015 included as the baseline year in the LEAP modeling. One of the main reasons for Sierra Leone's struggling

electricity system is that little attention has been given to forecasting electricity demand and supply. In 2015, the GoSL, with support from the World Bank through the MoE, contracted the National Rural Electric Cooperation Association (NRECA), a US electric cooperative, to prepare a transmission and distribution investment plan that the government will use to expand reliable electricity services to the capital, Freetown, and its surroundings [37]. However, due to government financial constraints, most of the priorities and recommendations provided by NRECA are yet to be implemented. This work developed three supply-side scenarios for determining the country's demand and supply forecasts. The three scenarios describe potential energy demand and how it will evolve in the future under various anticipated conditions. These scenarios include Base, Middle, and High cases, with the Middle and High cases incorporating more renewable (hydro and solar) and thermal LNG power plants into the system. The input data into the LEAP program includes various key assumptions, as well as demand, transformation, and resources. The key assumptions module consists of Gross Domestic Product (GDP), GDP growth rate, population, population growth rate, number of consumers, and consumer growth rate. Households, and the industrial, commercial, agricultural sectors, etc., constitute the demand sector. The transformation sector consists of energy transformed from energy sources as input into energy products using a range of electricity generation technologies. These technologies include those currently in operation during the base year and other proposed generations by the government. They also include emerging trends in generation technologies in the future. Dispatch rules, process efficiency, historical production, exogenous capacity, maximum availability, capital cost, and fixed and variable O&M costs are considered for each technology. Figure 4 shows the electricity generation system framework for Sierra Leone.

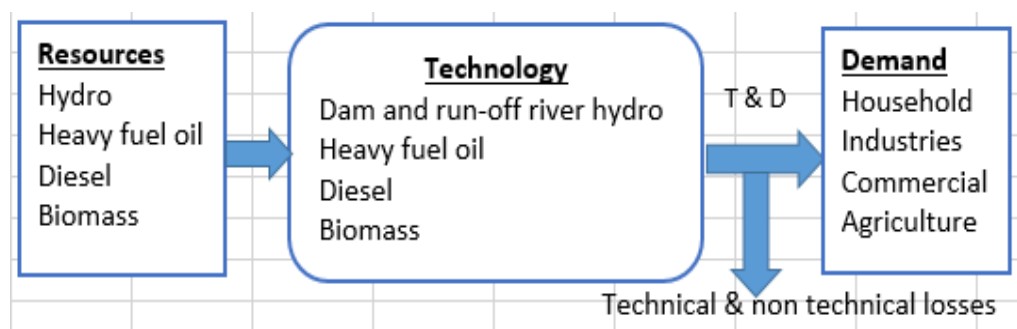

**Figure 4.** Electricity Generation system framework for Sierra Leone.

*3.2. Scenario Assumptions for the Sierra Leone LEAP Model*

As a result of the electricity consumption per capita over the past few years as well as the projected growth rate, the electricity demand projections for each scenario are entirely different. Therefore, the High scenario has higher demand projections, followed by the Middle and Base scenarios. These differences in demand projections will allow us to model different levels of demand. In addition, they will allow us to identify and evaluate the behavior of the future structure of the power generation system. Tables 5 and 6 show a summary of the data for the base year utilized in the LEAP Model of this work. They also show the input data for the generation technology. These data were sourced from MoE in hard copies. However, studies conducted in [38] indicate that, by 2035, electricity access in the country is expected to be 100%. Over the past years, electricity generated in the country has been mainly consumed by households, the production industry, and commercial. The mining industry and agricultural sector rely on self-generated electricity, mainly due to limited generation, transmission, and distribution lines [39]. Figure 5 summarizes electricity demand for various entities from 2015 to 2021.

**Table 5.** Model input parameters in the scenarios for the baseline year.

| Parameters | 2019 | 2040 | Comments |
|---|---|---|---|
| Population (Million) | 7.5 | 10.35 | Growth rate 2.54% SSL |
| GDP (Billions USD) | 4.22 | 9.69 | Growth rate 6.7% GoSL |
| Electricity access rate (%) | 26 | 97 | Electrification target (GoSL) |
| Electricity access rate urban | 38.9 | 98 | Electrification target (GoSL) |
| Electricity access rate rural | 2 | 56 | Electrification target (GoSL) |
| Electricity consumption per capita (kWh) | 34 | 250 | |
| T & D Losses (%) | 38 | 9 | |

**Table 6.** Model input data for generation technology [40–42].

| Parameters | Hydro | Oil (Thermal) | Solar | LNG |
|---|---|---|---|---|
| Lifetime | 30 | 30 | 30 | 30 |
| Merit order | 1 | 2 | 1 | 2 |
| Capital cost (USD/MW) | 3,100,000 | 1,290,000 | 2,100,000 | 2,600,000 |
| Fixed O & M cost (USD/MW) | 36,810 | 43,780 | 33,670 | 43,780 |
| Variable O & M cost (USD/MW) | 1.46 | 5.96 | 0.0 | 5.96 |
| Fuel cost (USD/Metric ton) | | 1282.35 | | 0.18 |

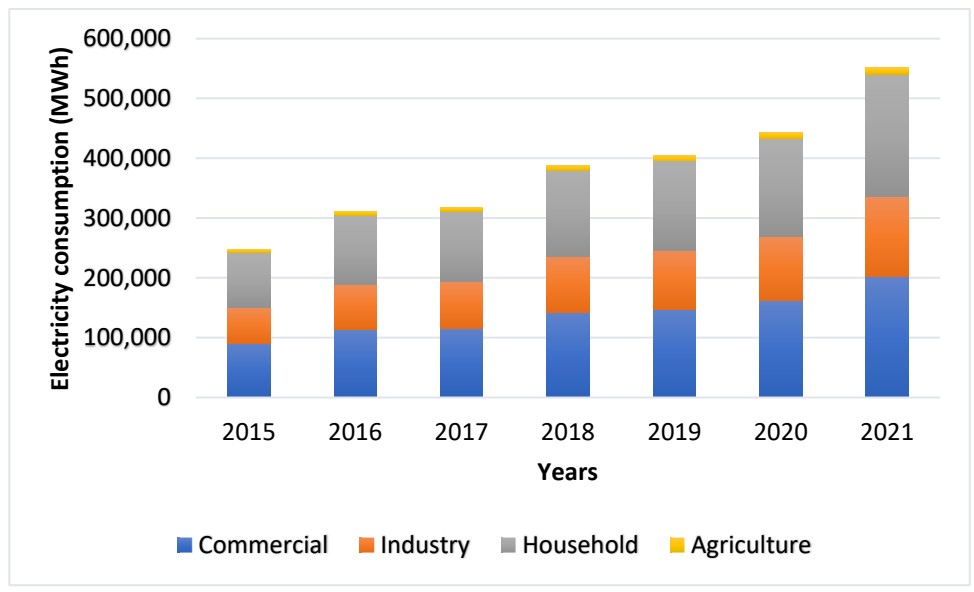

**Figure 5.** Electricity consumption by various entities from 2015 to 2021.

*3.3. Scenario Development*

Scenarios are crucial to the sustainable development of the energy sector. The future of energy has many factors: technological improvement, environmental regulation, variation in energy prices, the impact of climate change, and political changes, to name only a few of the areas of uncertainty [43]. This work implemented three scenarios, Base, Middle, and High cases, that represent the guideposts that can help us make decisions and take actions in the short, medium, and long term, and to fully understand the uncertainties over the entire range of possible outcomes. In the Base case, this work looks at the current status

of the power system in the country. The Middle case looks at the Sierra Leone electricity sector roadmap, while the High case looks at the implementation and sustainability of the electricity sector roadmap. Table 7 illustrates the three cases' parameters, considerations, and assumptions.

**Table 7.** Conditions for the alternative scenarios (Base, Middle, and High cases).

| Socio-Economic Drivers | Base | Middle | High |
|---|---|---|---|
| Household growth rate (%) | 2.24 | 2.54 | 3.54 |
| Growth rate of GDP (%) | 5.8 | 6.0 | 6.7 |
| Electricity demand growth rate (%) | 3.5 | 4.8 | 6.3 |
| T & D Losses (%) | 38 | 23 | 9 |
| Penetration of LPG stove (%) | 1 | 15 | 25 |
| Technology development drivers | Base | Middle | High |
| High penetration of energy-efficient appliances | yes | yes | yes |
| Ban on incandescent bulbs by 2030 | yes | yes | yes |
| Energy Efficient measures in place in buildings | yes | yes | yes |
| Energy Efficient measures in place in industries | yes | yes | yes |

### 3.3.1. Base Case: Business as Usual

In this scenario, we analyze the behavior of the electricity sector in Sierra Leone from 2015 to 2021, and we assume that future development trends will follow past trends with no new policy measures. The country's energy generation is presently dominated by diesel and heavy fuel oil thermal plants [44]. In this scenario, current energy projects using hydro, solar, and LNG thermal plants [45–47] are considered, and it is expected that electricity generation technologies, efficiency, transmission and distribution losses, percentage share of electricity production, fuel use, installed plant capacity, electricity consumption growth, and economic growth will follow the same pattern.

### 3.3.2. Middle Case: Electricity Sector Reform Roadmap

This scenario considers the electricity sector reform roadmap and the project for enhancing the Sierra Leone electricity access project (ESLEAP), which supports government electrification goals as stipulated in the medium-term national development plan (MTNDP) (2019–2023) [48,49]. One of the priorities of the ESRR is to identify the most relevant actions required in the short, medium, and long term to successfully develop the electricity sector and expand electricity generation and access to support government policy goals [50]. ESLEA aims at increasing the country's electricity access from about 15% to 30%, including the electrification of all district headquarters towns by 2023 [51]. The ESLEAP project directly supports three key energy sector objectives of the Government, as outlined in the MTNDP:

1. electrification of all district headquarters, which is a critical element of the country's electrification strategy;
2. the increase of rural electrification through engagement and involvement of key stakeholders, including the private sector;
3. improvement of sector financial performance for sustainable development [49].

This work incorporates the current solar hybrid project with a battery storage system under construction under the seven-district electrification project by the GoSL, which is expected to be completed by 2023. The United States International Development Finance Corporation (DFC) has also committed to providing USD 217 million in debt financing for a new 83-megawatt power plant to be installed in Freetown between 2023 and 2024. The project will be sponsored by Milele Energy and TCQ Power Limited and will generate and sell electricity to EDSA pursuant to a 20-year power purchase agreement [45]. A 50 MW solar power plant will be installed between 2023 and 2024 under the International Finance Corporation (IFC), a corporate subsidiary of the World Bank [52]. Three more hydroelectric

power stations will be installed and operated between 2025 and 2030. These will include 50 MW in the northern part of the country, 143 MW in the eastern part of the country, and 10 MW in the south of the country [52–54].

### 3.3.3. High Case: Sustainable Power Generation

In this scenario, we look at how to improve the current power generation system, which primarily utilizes fossil fuel oil, to a more sustainable system, composed primarily of renewable sources and producing less $CO_2$ emissions. The concept of this pathway is to reduce the share of fossil fuels and inject a greater share of renewable energy while also reducing the share of emitting power plants (LNG thermal plants) into Sierra Leone's overall energy mix for sustainable and economic power generation. Currently, the country's electricity generation is primarily derived from IPP, which is not sustainable and, at the same time, not economical. Renewable energy options provide electricity with many advantages. It is free of GHG emissions, reduces fuel import costs, and increases power security. This scenario also corresponds to the electricity sector reform roadmap, in which energy policy decisions are considered for sustainable universal electricity generation.

### 3.4. Demand Projection

Figure 6 illustrates the projection for the cumulative electricity demand for the three case scenarios generated. The forecasting is performed up to 2040. The projected electricity demand for the Base case grew from 791.1 GWh in 2019 to 1812.5 GWh in 2040, representing an increase of 129.11%. The Middle case electricity demand increased to 1936 GWh in 2040 from 791.1 GWh in 2019, resulting in an increase of 144.72%. The projected electricity demand for the High case increased to 2635.8 GWh in 2040 from 791.1 GWh in 2019, representing an increase of 233.18%. The most significant electricity demand in the Base case is the household sector, followed by commercial, and then by the industrial and agricultural sectors. Figure 7 shows the sector-wise electricity demand for the Base case. The electricity consumption growth rate for the various sectors from 2019 to 2040 indicates that households consume 47% of the electricity generated. This was followed by commercial at 23%, and industrial at 30%. Currently, actual electricity demand data for agriculture is not available. In terms of household energy use by fuel, charcoal use has increased gradually, in keeping with its trend in recent years. This has helped to reduce the substantial increase in household energy demand [55]. By 2040, the demand for electricity in the Base case is estimated at 89.2% of the total industrial and household sector demand, followed by charcoal which accounts for 5.4%. Figure 8 shows the energy demand by fuel type.

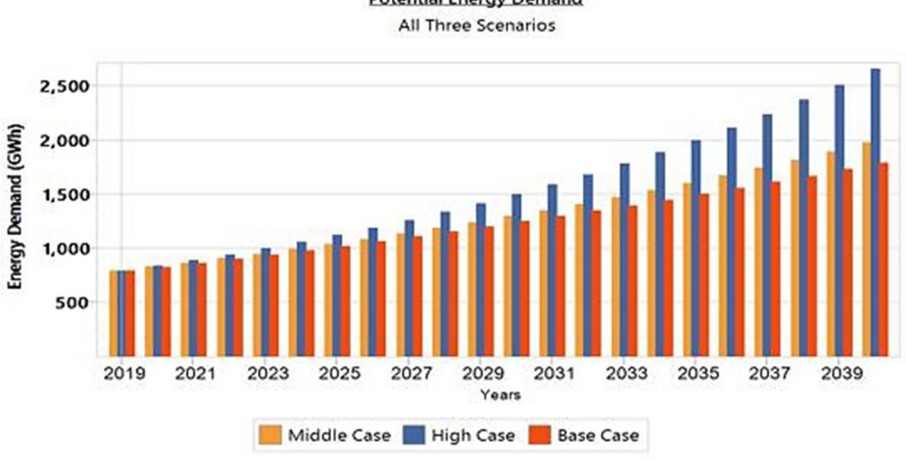

**Figure 6.** Projected Electricity Demand for Base, Middle and High Cases.

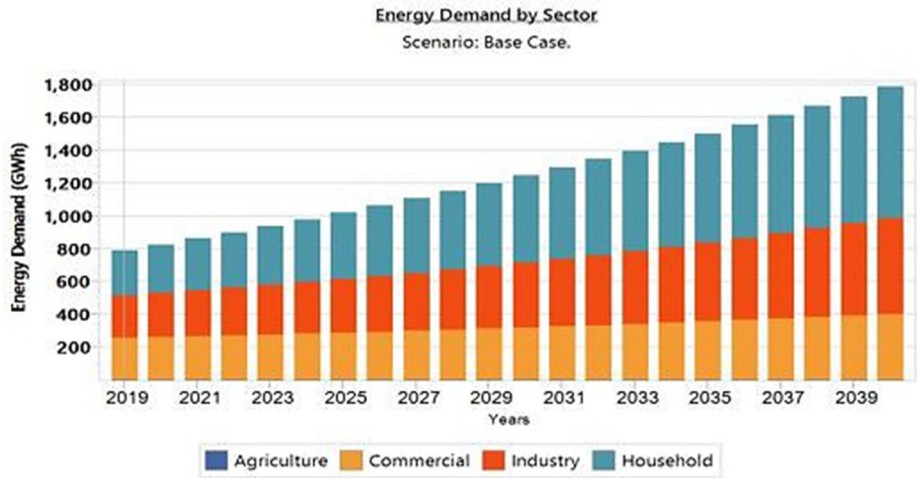

**Figure 7.** Sector-wise Electricity Demand—Base case.

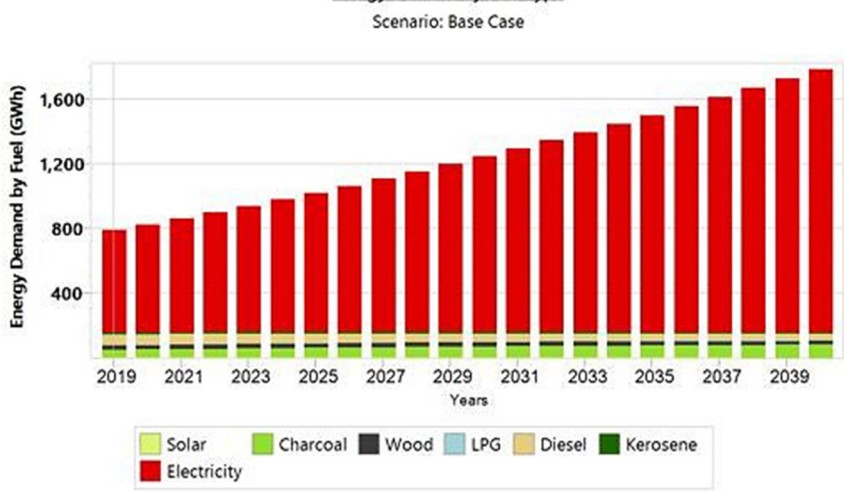

**Figure 8.** Outlook of Energy Demand by Fuel Type.

### 3.5. Supply Orojection

The supply projections look at the electricity generation that will meet the demand, as well as the installed capacity required. Model output for the three scenarios is projected using a variety of fuel and technology mixes to meet demand during the study period. Figure 9a–c shows electricity generation under all case scenarios of this study. The projected growth in electricity generation in 2040 from 2019 for the Base case is estimated to move from 441.9 GWh to 1812.5 GWh, representing an increase of 310.16%. The Medium case is estimated to reach 1936 GWh, while the High case claims 2635.8 GWh by 2040. This represents increases of 338.12% and 496.47%, respectively. By 2040, approximately 80% of the total electricity generation will come from renewable (Hydro, solar, and biomass). LNG and heavy fuel oil thermal plants will make up the remaining 20%. Current electricity generation projects in the country are predominantly based on hydro, solar, and LNG thermal plants. This study expects that, by 2040, all mining companies currently relying on their private generators for their sources of electricity would have been connected to the national grid. The projected installed capacity from 2019 to 2040 is estimated to move from 287.8 MW to 520.8 MW for the Base, to 525.1 MW for the Middle cases, and to 753.8 MW for the High case, as illustrated in Figure 10. The statistics indicate an increase of 80.9% for the Base case, 82.45% for the Middle case, and 161.9% for the High case. Approximately 80% of electricity generation in the High case will come from renewable sources. Therefore, the High case would be an ideal alternative scenario, considering that

the power generation system source is primarily composed of renewable sources that are sustainable, cost-effective, and environmentally friendly. Figure 11 shows installed capacity by source in High Case. A similar trend also took place in the Middle and Base cases.

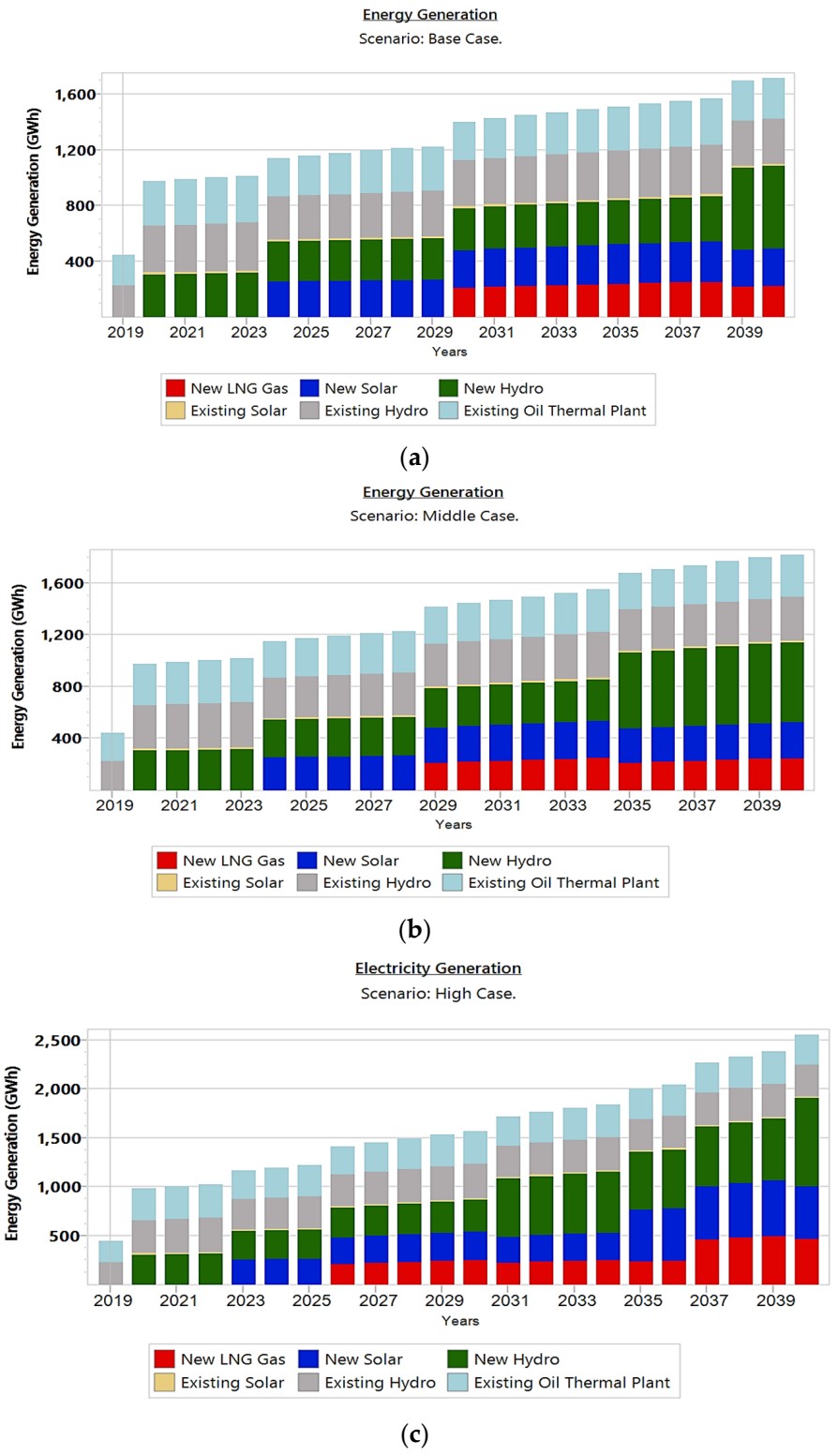

**Figure 9.** (**a**) Electricity Generation (Base case), (**b**) Electricity Generation (Middle case), and (**c**) Electricity Generation (High case).

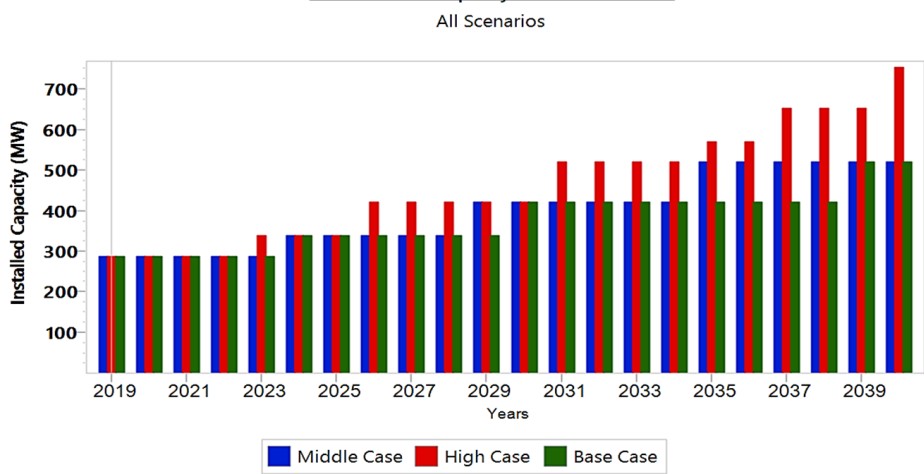

**Figure 10.** Installed Capacity for Base, Middle, and High cases.

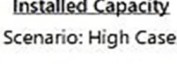
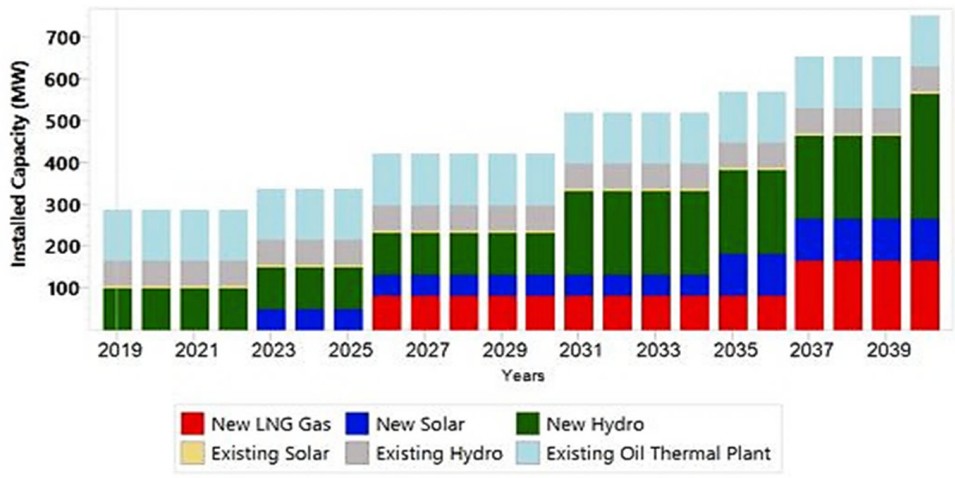

**Figure 11.** Installed Capacity by Source (High case).

## 4. Cost Analysis

The social cost was analyzed in terms of NPV, which included the annualized capital cost, and fixed and variable O and M costs at an assumed discounted rate of 5% during the study period for each case scenario, as indicated in Figure 12. The cumulative cost of production for the three cases involves capital, fixed and variable O and M, and fuel costs, as illustrated in Figure 13. In our analysis, we found that, from 2019 to 2040, the Base case had the highest production costs. This was followed by the Middle and High cases, with estimated values of USD 9217.9 million, USD 3961.4 million, and USD 3028 million, respectively. The high cost of production in the Base case was a result of the substantial involvement of fuel oil thermal plants in energy generation. Consequently, the cost of purchasing fuel led to a high production cost in the Base case. However, in the Middle and High cases, the production cost was less than the Base case. This was because these case scenarios utilized more renewable energy sources for power generation, accounting for approximately 60% and 80%, respectively. Figure 14 illustrates the net present capital value for the Base, Middle, and High cases from 2019 to 2040. The capital cost for the High case was higher than that of the Middle case, with an estimated value of USD 1186.8 million. The capital cost for the Middle case was greater than that of the Base case, with estimated values of USD 777.5 million and USD 659.4 million, respectively. The difference in cost

was because more renewable sources were used for energy generation in the Middle and High cases. In addition, renewable energy plants have higher capital costs than fuel oil plants. Because of the introduction of more renewable energy in the Middle and High cases, the high capital cost in these cases was neutralized in the production costs. This was because renewable energy is naturally produced and does not require heavy fuel oil for energy generation. The fuel cost for the High and Middle cases, with reference to the Base case, was negative, with a value of negative USD 6503 million and USD 5136.6 million, respectively, as indicated in Figure 15.

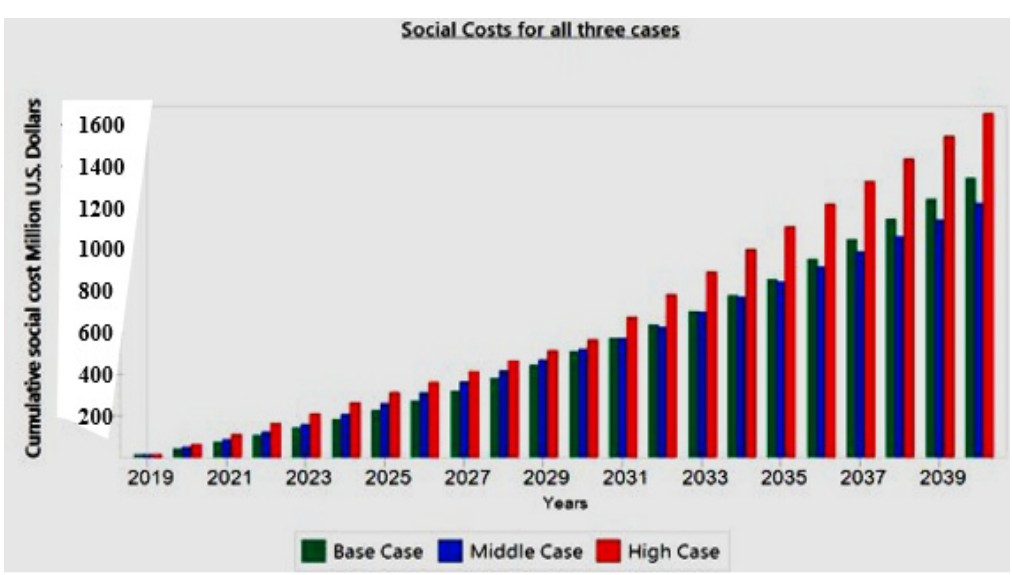

**Figure 12.** Social costs for Base, Middle, and High case scenarios.

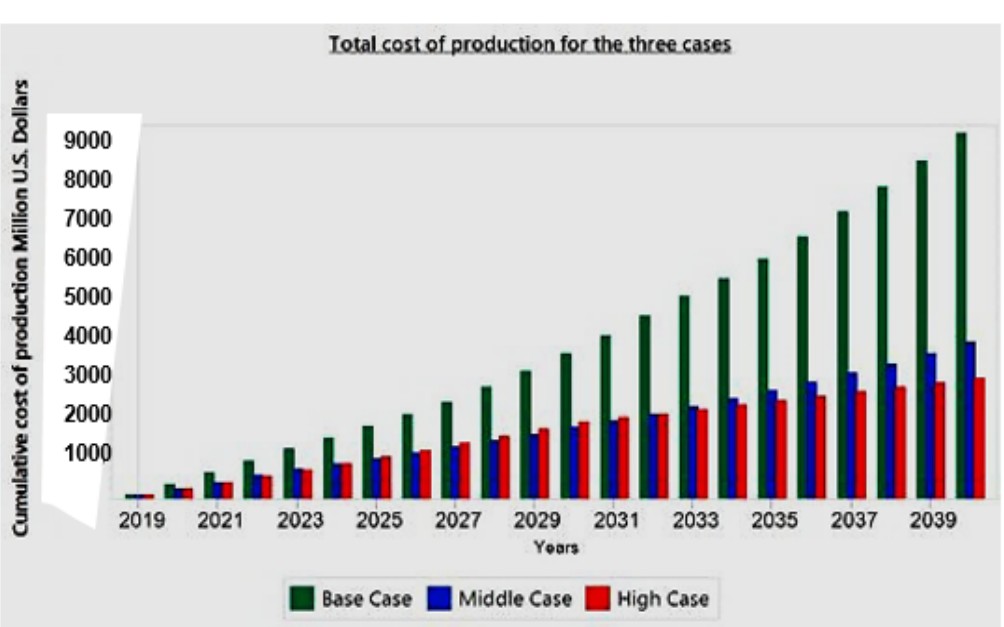

**Figure 13.** Cumulative cost of production for Base, Middle, and High case scenarios.

*4.1. Sensitivity Analysis*

The social cost for the system was expressed in terms of net present value (NPV) discounted at 3%, 5%, 7%, 8%, and 10%, as shown in Figure 16. The volatility of fossil fuels and the high emission of greenhouse gases has made the Middle and High cases, with high penetration of renewables, more attractive than the Base case. The High case scenario was

estimated to have the lowest NPV when compared to the Middle and Base case scenarios at all discount rates. Considering the NPV at the discount rate of 5%, which was estimated to be the GoSL policy rate, the High case scenario had the lowest value. This was as a result of a significant reduction in HFO generation plants. The total savings at a discount rate of 5% for the High case scenario was estimated at USD 2011.90 million, compared to the NPV of the Middle case, which was USD 2388.95 million, and the NPV of the Base case, which was USD 5190.9 million savings. Looking at the NPV of all three case scenarios, the High case scenario had the least-cost electricity generation pathway at all discount rates. The Middle case scenario had a lower NPV than the Base case. The NPV at all discount rates for the Base case scenario was higher than all other scenarios. This was due to the involvement of a greater number of conventional power generation technologies.

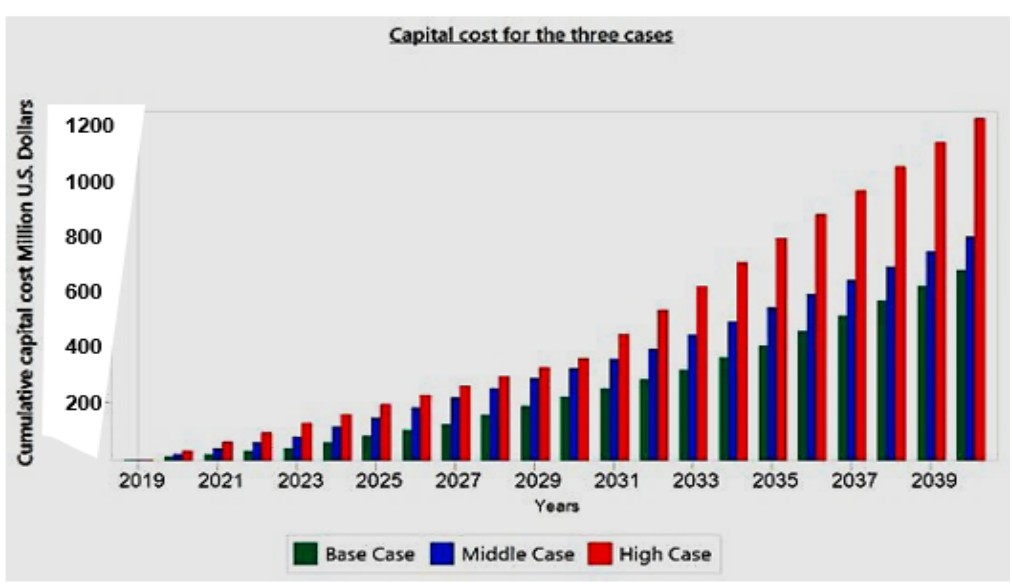

**Figure 14.** Cumulative capital cost for Base, Middle, and High case scenarios.

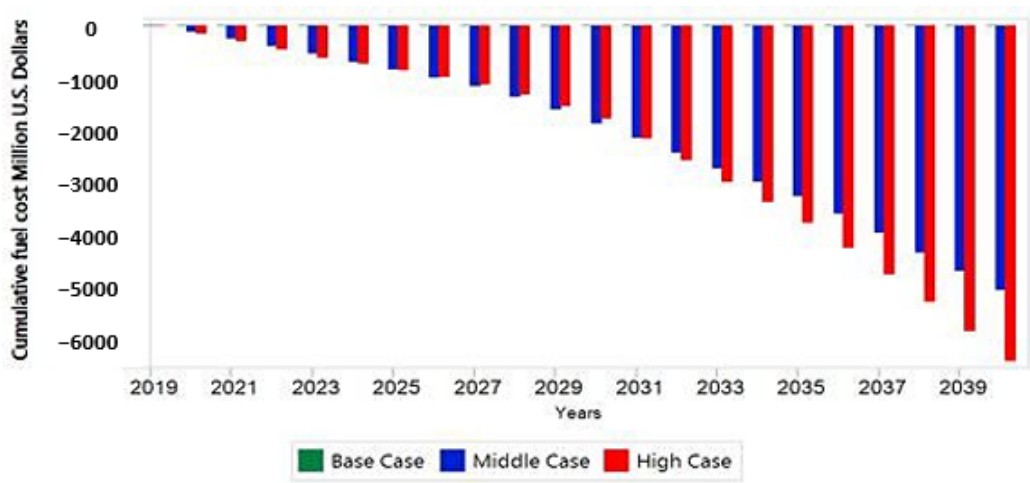

**Figure 15.** Fuel cost for the High and Middle cases with reference to the Base case.

### 4.2. Greenhouse Gas Emission

The combined carbon dioxide emissions from the Base case's energy demand and supply sectors are estimated to reach 16.2 Mt by 2040; 97.1% of this $CO_2$ emission was estimated to come from the supply side, while the remaining 2.9% was expected to come from the demand side. This is indicated in Figure 17. The large and extensive use of heavy fuel oil combustion engines for energy generation in the Base case was seen as the

primary producer of carbon dioxide by 2040, responsible for about 97.1% of the emission, followed by biomass in the form of firewood and charcoal for cooking with 2.3%. Kerosene was said to account for approximately 0.1%, while natural gas (LPG) accounted for the remaining 0.5%. Figure 18 illustrates carbon dioxide emission by fuel category for the Base case. Figure 19 shows the carbon dioxide emissions for the three case scenarios. The Base case had the highest carbon dioxide emissions, approximately 16.2 Mt, by 2040. This was mainly due to the high penetration of heavy fuel oil thermal plants for power generation in the Base case. The High case showed the lowest $CO_2$ emissions, about 7.8 Mt, by 2040, accounting for almost half of the $CO_2$ emission in the Base case. The low emission of $CO_2$ in the High case is due to the significant penetration of renewable energy into the power system. $CO_2$ emissions in the Middle case were significantly lower than in the Base case, at about 8.6 Mt. Also, in the Middle case, there is more penetration of renewable energy sources, as most renewable energy projects are expected to start in the Middle case.

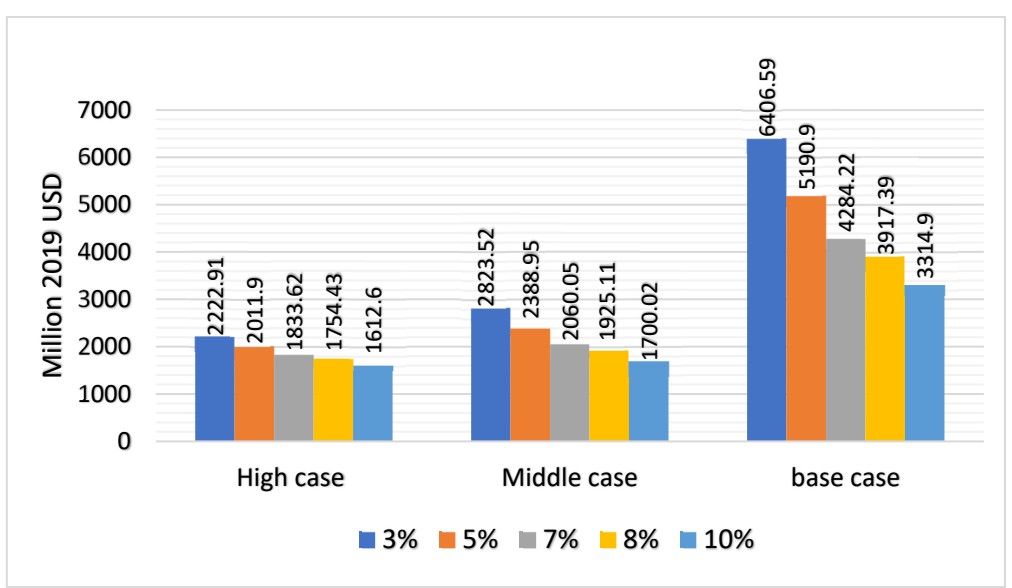

**Figure 16.** NPV at different discount rates.

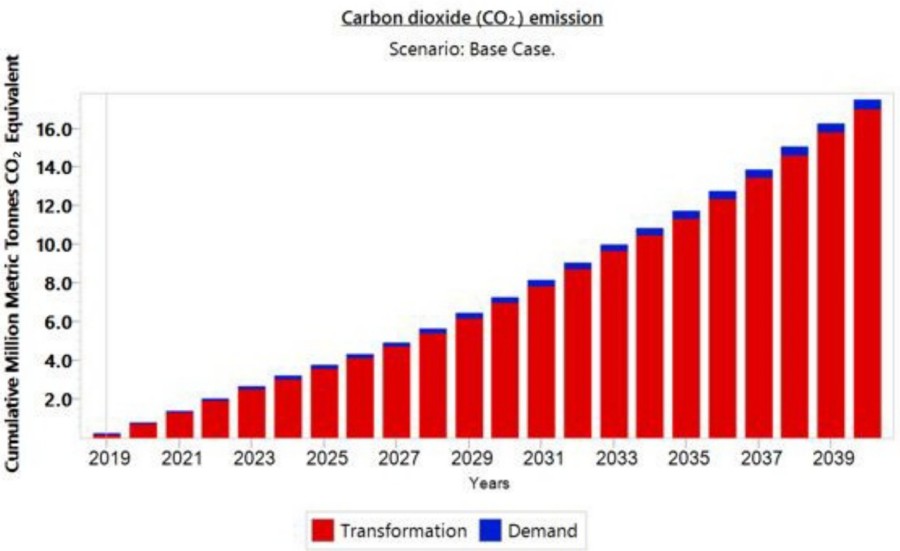

**Figure 17.** Carbon dioxide emissions for transformation and demand side (Base case).

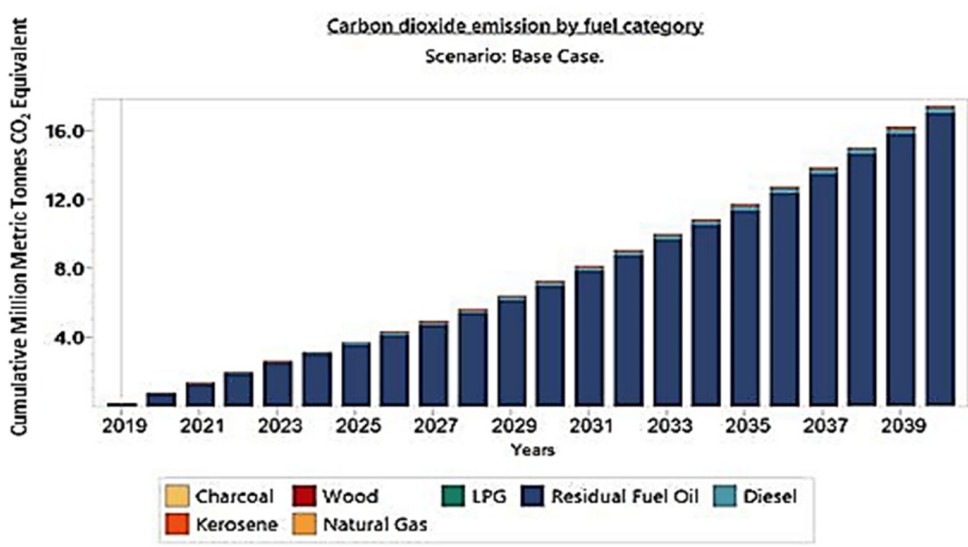

**Figure 18.** Carbon dioxide emissions by fuel type (Base case).

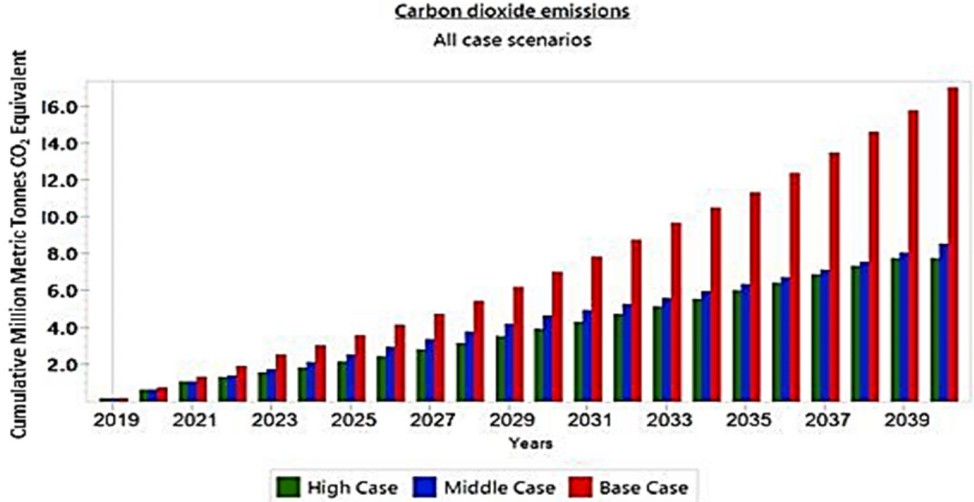

**Figure 19.** Carbon dioxide emissions for High, Medium and Base cases.

## 5. Discussion

In this study, an integrated resource planning approach is proposed to assess the country's current and future electricity supply and demand. The objective is to ensure that generation meets demand, enhances reliability, minimizes the total cost, and reduces environmental costs and greenhouse gas emissions. In general, three different scenarios were developed for addressing current and future electricity demand using LEAP software. To simulate demand forecasts from 2019 to 2040, three possible scenarios were used: Base case, Middle case, and High case. Final energy consumption was modeled using demand analysis, with demand modules for household, industrial, commercial, and agricultural sectors. Looking at the cumulative production costs for the final energy consumed, our results indicate that the High case may have a significant adverse effect over the Middle and Base cases. The initial investment in renewable energy is expensive. That is why, in our study, our results showed that the High case had the highest capital cost when compared to the Middle and Base cases. However, due to the large penetration of renewables in the High case, our results indicate that the High case may be more viable with respect to the Middle and Base cases in terms of fuel consumption for power generation.

In general, our study revealed that the High case scenario may be the least-cost electricity generation pathway. In addition, our study also indicates that the high case may

have the least greenhouse gas emission, thereby contributing greatly to reducing global warming. The demand forecast from our study shows that the projected values of electricity demand for the generated case scenarios from 2019 to 2040 are 1812.5 GWh for the Base case, 1936 GWh for the Middle case, and 2635.8 GWh for the High case. The projected installed capacity is expected to be 520 MW for the Base case, 525.1 MW for the Middle case, and 753.8 MW for the High case. Therefore, our study results indicate that the High case may be the most suitable option for meeting the electricity demand for 2040, considering the current trend of electricity consumption. However, in all three case scenarios, a substantial amount of generation capacity is needed in the year 2040 in order to meet the country's electricity demand.

In 2040, cumulative $CO_2$ emissions are estimated to increase from 0.2 million Mt in 2019 to 16.2 million Mt in the Base case, 8.6 Mt in the Base case, and 7.8 Mt in the High case. $CO_2$ emissions in the Base case are high primarily due to the high use of conventional thermal technology. According to the projections, the total cumulative production cost of electricity generation in the year 2040 for Base, Middle, and High scenarios is USD 9217.9 million, USD 42,411.4 million, and USD 3971.2 million, respectively, discounted to the value of 2019. Air pollution levels can be reduced if renewable energy is given a prominent place in the state's energy plan. One of the scopes of the electricity sector roadmap was to create an effective environmental management system that protects biodiversity and preempts environmental disasters as well as being a model for responsible and efficient exploitation of natural resources. Under the reform path, the MoE developed an energy policy update that reconciled and simplified the various reference policy documents for the energy sector, including the Renewable Energy Policy 2015. The Renewable Energy Section of the Energy Policy Update called for the development of a Feed-in-Tariff Policy as a policy instrument to support renewable energy. The incorporation of renewable energy into the power system will lead to a substantial reduction in production cost, GHG emission, and will enhance economic development.

Using the Long-range Energy Alternative Planning (LEAP) model, Emodi et al. [56] explore Nigeria's future energy demand, supply, and associated GHG emissions from 2010 to 2040. In this study, four scenarios were considered: the reference scenario, the low-carbon moderate scenario, the low-carbon advanced scenario, and the green optimistic scenario. According to their study, the REF scenario led to a demand of 3075PJ in 2040, from 1039PJ in 2010. As a result, GHG emissions will rise from 50.2 Mt in 2010 to 201.2 Mt in 2040. The rise in GHG emissions was a result of the extensive use of fossil fuel oil for energy generation. Regarding the alternate scenarios, demand for 2040 is expected to drop to 2941PJ for LCW, 2488PJ for LCA, and 2249PJ for GO scenarios. The level of GHG emissions for 2040 will also drop to 185 Mt for LCM, 152 Mt for LCA, and 124.4 Mt for GO. The low energy demand and GHG emissions in the three scenarios resulted from energy efficiency programs, the implementation of aggressive energy policies such as the introduction of LPG in the power system, and the high level of renewable energy applications such as solar power. Although the Nigerian power system is larger than the Sierra Leone power system, this confirms our finding that the extensive utilization of renewable power in the power system will not only reduce production costs but also reduce the level of GHG emission into the air, thereby contributing to minimizing global warming. We can also see the demand cost for LCM, LCA, and GO relative to REF scenarios are USD 36.62, 31.32, and 30.61 million, respectively. Rivera-González et al. [42] assess the Ecuadorian power generation system, estimating the electricity supply and demand forecast until 2040 by using the Long-range Energy Alternative Planning (LEAP) model. The author proposes three different scenarios: Business as Usual (S1), Power Generation Master Plan (S2), and Sustainable Power Generation System (S3). The study results indicate that the net electricity demand for S1 increased from 20,204 GWh in 2017 to 44,847 GWh in 2040, that for S2 it increased from 20,204 in 2017 to 64,728 GWh in 2040, and that for S3 it increased from 20,204 GWh in 2017 to 72,053 GWh in 2040. The average electricity production cost in S1 is higher than the S2 and S3 scenarios during the total analysis period, with S1 showing

USD 19.52/MWh, S2 USD 17.48/MWh, and S3 USD 17.78/MWh. The reason for the lower production costs in S2 and S3 is as a result of the substitution of petroleum fuels by natural gas and the reduction in the use of petroleum fuel oil. Furthermore, results indicated that fuel consumption in S2 and S3 greatly reduced when compared to S1. The reduction in fuel consumption in S2 and S3 is due to the increased use of natural gas, bagasse, and biogas when compared to S1, which is dominated by diesel, residual fuel oil, and crude oil. These results also confirm our study that the increased use of renewable energy in the power system will increase economic growth through a reduction in production costs and can also contribute to the fight against climate change. Kale et al. [18] propose an assessment of the electricity demand and supply scenarios for Maharashtra (India) using the LEAP model. The author generated three scenarios in this study: Business as Usual (BAU), Energy Conservation (EC), and Renewable Energy (REN). Results from the study showed that total electricity generation for BAU moved from 78.4 billion kWh in 2012 to 253.2 billion kWh in 2030, 78.4 billion kWh in 2012 to 188.4 billion kWh in 2030 for the EC scenario, and 78.4 billion kWh in 2012 to 253.2 billion kWh in 2030 for the REN scenario. The decrease in generation for the EC scenario is due to the implementation of an energy conservation policy. However, GHG emissions for the BAU scenario increased from 65.7 Mt in 2012 to 226.4 Mt in 2030, and for the EC scenario, it moves from 65.7 Mt in 2012 to 165.8 Mt in 2030; the REN scenario experienced a significant reduction of GHG emissions, from 65.7 Mt in 2012 to 35.5 Mt in 2030. The significant rise in $CO_2$ emissions in the BAU model is because the new power plants are mainly based on coal, while the implementation of energy conservation measures saves a large amount of capacity addition, thereby reducing the amount of GHG emissions. For the REN scenario, all the new generation plants, as well as replacements of the retired ones, were renewable-based power plants. Although the REN scenario has the highest capital cost, its total cost is the lowest, at approximately INR 5692.7 billion; costs were INR 8498.3 billion for the EC scenario and INR 10,247.2 billion for the BAU scenario. This is due to the use of renewable energy and energy conservation policy in the REN and EC scenarios, respectively. Ibrahim et al. [36] propose an analysis of the electricity demand and supply for Nigeria using the LEAP model. The model employs three scenarios (Business as usual (BAU), Energy Conservation (EC), and Renewable Energy (REN)) to ascertain the electricity supply and demand for the period from 2010 to 2040. The study showed that the amount of electricity needed for the BAU and REN scenarios increased from 35.9 billion kWh in 2010 to 283.6 billion kWh in 2040. The amount needed for the EC scenario increased from 35.9 billion kWh in 2010 to 233.8 billion kWh in 2040. In the BAU scenario, GHG emissions increased from 6 Mt to 123.9 Mt, and in the EC scenario, they increased from 6 Mt to 84.6 Mt. The REN scenario has the least GHG emissions. The significant rise in GHG emissions for the BAU is due to the addition of coal power plants and the expansion of the capacity of the gas power plants. Low emission of GHG in the EC scenario is a result of the energy efficiency measures taken, which led to a reduction in electricity demand and supply. The utilization of renewable energy sources in the REN scenario means it has the lowest GHG emissions.

Few studies have been conducted on Sierra Leone's energy demand forecast. A study conducted by the NRECA in 2015 indicated that the electricity demand for 2020 and 2030 was forecasted at 503 GWh and 1447 GWh, respectively [37]. However, the projected demand forecast may have been overemphasized. Under the Sierra Leone electricity sector reform roadmap for 2017 to 2030, prepared by MoE with support from MCCU, the projected power demand forecast for 2030 was 1800 MW [45]. The 2022 energy sector roundtable, presented by MoE, shows an estimated demand of 850 MW by 2030. However, the above studies may not be detailed. These studies were found to focus on the medium-term goal. In addition, the demand forecast for the year 2030 may have been overemphasized considering the current power demand in the country. With sufficient statistical data, our study carefully looked at the power demand trend and approximated the long-term power demand in the country.

The significant involvement of conventional generation technologies in the Base case will have to be abandoned. This is due to the numerous constraints faced by the government in terms of fuel purchasing, frequent procurement of spare parts, and its adverse effect on the environment. Despite the high capital cost of renewable energy sources, its advantages include adding capacity at low O and M costs and having high efficiency with no fossil fuel oil cost. Therefore, all future energy projects by the government will be based on renewable energy. It is likely that these technologies will be installed in strategic locations with high electricity demand, such as mining areas and areas where the extension of the grid may be difficult and expensive. These potential locations may also be close to the CLSG 225 kV transmission line so that the generated power could be easily transported to load centers using this line.

Sierra Leone is yet to produce its nationally appropriate mitigation action (NAMA). The country's global historical contribution to greenhouse gas emissions is estimated to be 0.1%. The country's projected total emissions from all sources and sectors and for all gases are projected to increase from about 4.8 Mt $CO_2$ equivalent in 2015 to about 6.6 Mt $CO_2$ equivalent in 2030 [57]. Sierra Leone's intended nationally determined contribution (INDC) is framed in terms of its desired outcomes. Through this INDC, the country is committed to implementing specific emissions-reduction actions, such as policies or mitigation actions including advancing a feed-in tariff for renewable energy technologies, phasing out fossil fuel subsidies, or converting to no-tillage agricultural practices. Our study also indicates that a substantial $CO_2$ emission reduction by 2040 may also help meet Sierra Leone's intended nationally determined contribution (INDC), committed to implementing specific emissions-reduction actions in line with the Paris Agreement in achieving Sustainable Development Goals (SDGs). We believe that this may serve as a reference to the government of Sierra Leone for mapping out strategies for addressing energy demand by 2040. Furthermore, this work can be further expanded by incorporating energy efficiency and energy management strategies. Finally, this work proposes a clear pathway for meeting the electricity demand by 2040 in a more cost-effective, reliable, sustainable, and carbon-free manner, as indicated in the Sierra Leone electricity roadmap.

## 6. Conclusions

This study focuses on forecasting the long-term electricity demand-supply situation in Sierra Leone by considering techno-economic and environmental parameters. Three case scenarios have been generated (Base, Middle, and High) that will cover the country's total electricity demand. The three case scenarios were generated using the LEAP model, which describes the possible future state of the electricity demand-supply situation of Sierra Leone. Each scenario tried to present an image of what the future can hold for Sierra Leone's power sector in terms of generation, installed capacity, cost of production, and $CO_2$ emissions. The Base case, which is also business as usual, looks at the current energy settings, which are predominantly fuel oil thermal plants that are significantly dependent on petroleum products that the country imports (mainly diesel and heavy fuel oil). In the Middle and High cases, we examined the electricity sector reform roadmap and the sustainability of the electricity sector reform roadmap, respectively. The total electricity demand is expected to be covered mainly by hydroelectric power, solar, and LNG power plants. In the High case, the projected electricity demand for 2040 has been forecasted to be 2635.8 GWh, from 791 GWh in 2019, a growth 3.4 times higher than the base year demand, with an estimated annual average growth of 5.9% over the study period. In the Middle case, the projected electricity demand for 2040 was 1936 GWh, from 791 GWh in 2019, a growth 2.4 times higher than the base year demand, with an estimated annual average growth of 4.35% over the study period. Furthermore, in the Base case, the projected electricity demand for 2040 has been forecasted to be 1812.5 GWh, from 791 GWh in 2019, a growth 2.3 times higher than the base year demand, with an estimated annual average growth of 4.02% over the study period. However, in the Medium and High cases, more renewables are brought into the power system, which reduces the production costs significantly due to

less involvement of fuel oil thermal plants. In that case, the High case is said to have the lowest production cost, followed by the Middle case, with the Base case having the highest. In addition, the High case also has the least $CO_2$ emissions, followed by the Middle and the Base cases. Therefore, for meeting energy demand by 2040, the High and Middle cases are the most economical, environmentally friendly, and sustainable options. The cost-benefit analysis has drawn comparisons between the scenarios based on a cumulative total cost discounted rate of 5%. The Base case is the most expensive regarding the production cost, while the High case is the most economical, since the fuel cost forms a significant chunk of the production cost. The social cost for the High case is higher than the Base case because of the high penetration of renewable energy in the High case. In terms of cost, the High case is preferred to the Middle case, and the Middle case is preferred to the Base case. The sensitivity analysis shows that the increase in fuel costs and high $CO_2$ emissions make the High and Middle cases more attractive than the Base case. The ongoing decrease in the cost of renewable technology has made the High and Middle cases more attractive. Hence, the proposed model can be used as a planning tool by energy planners and government policymakers to construct new power generation system infrastructures.

**Author Contributions:** Conceptualization, F.C. and T.S.; methodology, F.C. and M.F.; software, F.C. and M.F.; validation, F.C.; formal analysis, M.F.; investigation, F.C.; resources, M.A.C.; data curation, S.S.R.; writing—original draft preparation, F.C. and M.F.; writing—review and editing, E.R.C., A.R. and M.A.C.; visualization, A.R. and M.F.; supervision, T.S.; project administration, T.S.; funding acquisition, T.S. All authors have read and agreed to the published version of the manuscript.

**Funding:** This research received no external funding.

**Institutional Review Board Statement:** Not applicable.

**Informed Consent Statement:** Not applicable.

**Data Availability Statement:** Not applicable.

**Conflicts of Interest:** The authors declare no conflict of interest.

## Abbreviations

The following abbreviations are used in this manuscript:

| | |
|---|---|
| GDP | Gross Domestic Product |
| EGTC | Electricity Generation and Transmission Company |
| EDSA | Electricity Distribution and Supply Authority |
| WAPP | West Africa Power Pool |
| NRECA | National Rural Electric Cooperative Association |
| DFO | Diesel Fuel Oil |
| HFO | Heavy Fuel Oil |
| MCC | Millennium Challenge Corporation |
| MW | Mega Watts |
| kV | Kilo Volts |
| kWh | Kilo Watts Hour |
| MCCU | Millennium Challenge Coordinating Unit |
| LNG | Liquified Natural Gas |
| PV | Photovoltaic |
| GW | Giga Watts |
| DGs | Diesel Generators |
| CLSG | Cote d'Ivoire, Liberia, Sierra Leone and Guinea |
| NRECA | National Rural Electric Cooperative Association |
| GWh | Giga Watt Hour |
| O%M | Operation and Maintenance |
| Mt | Million metric tonnes |

| MTNDP | Medium-Term National Development Plan |
|-------|---------------------------------------|
| ESLEAP | Enhancing the Sierra Leone Electricity Access Project |
| ECREEE | Economic Community of West African states Centers for Renewable Energy Effficiency |
| UNSDG | United Nations Sustainable Development Goal |
| RES | Renewable Energy Sources |
| MG | Micro-grid |
| PSP | Reform Sector Project |
| SSL | Statistics Sierra Leone |
| ERSP | Electricicty Reform Sector Project |
| JICA | Japan International Cooperation Agency |
| MoE | Ministry of Energy |
| EBID | ECOWAS Bank for Investment and Development |
| T & D | Transmission & Distribution |
| IPP | Independent Power Producer |

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
