# Peer review of "Long-Term Forecast of Sierra Leone’s Energy Supply and Demand (2019–2040): A LEAP Model Application for Sustainable Power Generation System"

_sustainability, doi:10.3390/su151511838_

Round 1
Reviewer 1 Report
Dear Authors.
The manuscript shows important information on the situation of the electricity sector in Sierra Leone. It employs the LEAP model to perform a prospective, and define the future situation based on scenarios.
The paper has major flaws in the way it is written. There are very long sentences and paragraphs with many sentences, which makes it difficult to read.
Regarding the use of LEAP, it does not define which are the assumptions to formulate the scenarios, considering that it is a bottom-up planning technique. Nor is there information related to other models that were applied or a state of the art associated with planning.
From the scientific point of view, I do not find a contribution, i.e., the novelty of the research (or if the paper is a research) is not specified. From my point of view it is a report, which is valuable for the information noted.
The discussion is not in accordance with the scientific method, since the results obtained are not put in context with the results of other research.
For this reason I consider that this document is not suitable for publication in a scientific journal.
The grammar should be improved and the paragraphs and sentences should be better constructed.
Author Response
We appreciate you and the for your precious time in reviewing our manuscript and providing valuable comments. It was your valuable and insightful comments that led to possible improvements in the current version. The authors have carefully considered the comments and tried our best to address every one of them. We hope the manuscript, after careful revisions, meets your high standards. The authors welcome further constructive comments if any. Please see the attachment

Reviewer 2 Report
Please view the attachment!

Minor editing of English language required!
Author Response
We appreciate you and the for your precious time in reviewing our manuscript and providing valuable comments. It was your valuable and insightful comments that led to possible improvements in the current version. The authors have carefully considered the comments and tried our best to address every one of them. We hope the manuscript, after careful revisions, meets your high standards. The authors welcome further constructive comments if any. Please see the attachment.

Reviewer 3 Report

The submitted manuscript suffers from minor mistakes in writings and grammatically errors. Please go over the manuscript for other grammatical errors and correct all of them.
Author Response

(The authors gave the same response as above.)

Reviewer 4 Report
Overall, this article presents interesting findings. However, it appears that the authors did not give due attention to the submission process, as the manuscript lacks proofreading and contains numerous editing issues. Consequently, I find it challenging to read and comprehend.
To enhance the quality of the manuscript, I would like to provide the following suggestions and comments:
No. |
Line No. |
Comment |
1 |
// |
The authors should clearly discuss the scientific contributions and novelty of their research. This aspect is currently lacking in the paper. |
2 |
// |
The figures included in the manuscript are unclear. I recommend using high-resolution images, such as saving Excel figures in .svg format, to improve clarity and readability. |
3 |
37 |
The Introduction section is disorganized and excessively long. I suggest dividing it into several sub-sections to enhance its structure and readability.
|
4 |
51 |
There is a missing full stop between "implications" and "Approximately." |
5 |
83 |
a "(" is missing. |
6 |
92 |
The positions of the bullet points need to be adjusted for clarity. |
7 |
122 |
The presence of multiple instances of the subscript, such as "1" ,”144”,”244”, throughout the paper indicates consistency issues. Please address these problems. |
8 |
208 |
A section title is missing for Section 2. Please provide an appropriate title. |
9 |
208 |
Including a map would be beneficial in aiding the understanding of this section, particularly regarding the location and capacity. Currently, it is challenging to grasp these aspects. |
10 |
281 |
The citation in the equation requires adjustment. Please ensure proper formatting and citation style. |
11 |
368-373 |
The order of sections 4, 5, and 6 needs to be rearranged. |
Best regards,
Author Response

(The authors gave the same response as above.)

Round 2
Reviewer 1 Report
Dear authors.
I consider that the manuscript is more a report than a scientific article. Although there is various information from Sierra Leone that could help to understand the energy situation, I still can't find the news. The LEAP model has been used in several countries, regions and even worldwide, so in this case only the model is applied. My arguments are reinforced by the lack of discussion, that is, the results obtained with other investigations are not placed in context.
For this reason I consider that the manuscript cannot be published, in the version sent.
I have no observations.
Author Response
The authors would like to thank the reviewer for the valuable and insightful comments that led to possible improvements to our manuscript. The authors welcome further constructive comments if any.

Reviewer 3 Report
I agree with the revisions for the manuscript done by the authors. No more comments, the author has incorporated good changes in the manuscript.
Author Response
The authors would like to thank the reviewer for the valuable and insightful comments that led to possible improvements to our manuscript

Reviewer 4 Report
In regard to my comment No.9, I believe that creating a map independently should not pose a significant challenge for the authors.
Author Response

(The authors gave the same response as above.)

Round 3
Reviewer 1 Report
Dear Authors I have reviewed the manuscript and I believe this has an interesting information about Sierra Leone, I didn't find the improvement in the discussion. In the research article, the discussion puts the results of the research in context with other research. The research must explain the results and compare them with other information. For my criteria, if an article doesn't have this topic, it is incomplete. You can see in this section that there are no citations and this is not coomun in a research article.Moderate editing of English language required
Author Response
Thank you for taking the time to review our manuscript and provide valuable feedback. It was your valuable and insightful comments that led to improvements in the current version. The authors have carefully considered the comments and tried to address them. We hope the manuscript, after careful revisions, meets your high standards. The authors welcome further constructive comments.
